# Restriction-Assembly: A Solution to Construct Novel Adenovirus Vector

**DOI:** 10.3390/v14030546

**Published:** 2022-03-06

**Authors:** Xiaojuan Guo, Yangyang Sun, Juan Chen, Xiaohui Zou, Wenzhe Hou, Wenjie Tan, Tao Hung, Zhuozhuang Lu

**Affiliations:** 1NHC Key Laboratory of Medical Virology and Viral Diseases, National Institute for Viral Disease Control and Prevention, Chinese Center for Disease Control and Prevention, Beijing 100052, China; guoxj@ivdc.chinacdc.cn (X.G.); syymsw@163.com (Y.S.); chenj_btmc@163.com (J.C.); zouxh@ivdc.chinacdc.cn (X.Z.); houwz@ivdc.chinacdc.cn (W.H.); hongt@cae.cn (T.H.); 2School of Laboratory Medicine, Weifang Medical University, Weifang 261053, China; 3School of Public Health, Baotou Medical College, Inner Mongolia University of Science and Technology, Baotou 014040, China; 4Chinese Center for Disease Control and Prevention–Wuhan Institute of Virology, Chinese Academy of Sciences Joint Research Center for Emerging Infectious Diseases and Biosafety, Wuhan 430071, China

**Keywords:** adenovirus, vector, construct, Gibson assembly, restriction enzyme, infectious clone, modification

## Abstract

Gene therapy and vaccine development need more novel adenovirus vectors. Here, we attempt to provide strategies to construct adenovirus vectors based on restriction-assembly for researchers with little experience in this field. Restriction-assembly is a combined method of restriction digestion and Gibson assembly, by which the major part of the obtained plasmid comes from digested DNA fragments instead of PCR products. We demonstrated the capability of restriction-assembly in manipulating the genome of simian adenovirus 1 (SAdV-1) in this study. A PCR product of the plasmid backbone was combined with SAdV-1 genomic DNA to construct an infectious clone, plasmid pKSAV1, by Gibson assembly. Restriction-assembly was performed repeatedly in the steps of intermediate plasmid isolation, modification, and restoration. The generated adenoviral plasmid was linearized by restriction enzyme digestion and transfected into packaging 293 cells to rescue E3-deleted replication-competent SAdV1XE3-CGA virus. Interestingly, SAdV1XE3-CGA could propagate in human chronic myelogenous leukemia K562 cells. The E1 region was similarly modified to generate E1/E3-deleted replication-defective virus SAdV1-EG. SAdV1-EG had a moderate gene transfer ability to adherent mammalian cells, and it could efficiently transduce suspension cells when compared with the human adenovirus 5 control vector. Restriction-assembly is easy to use and can be performed without special experimental materials and instruments. It is highly effective with verifiable outcomes at each step. More importantly, restriction-assembly makes the established vector system modifiable, upgradable and under sustainable development, and it can serve as the instructive method or strategy for the synthetic biology of adenoviruses.

## 1. Introduction

Adenoviruses are non-enveloped viruses with an icosahedral nucleocapsid, which packs a genome of linear double-stranded DNA of 26−48 kb. Adenoviridae is divided into five genera, of which Masteradenovirus infects mammals [1]. Adenoviruses have been isolated from many mammals, such as humans, monkeys, bovines, and mice. Human adenoviruses contain more than 100 types (http://hadvwg.gmu.edu/, accessed on 5 February 2022), which belong to seven species (HAdV-A to -G) [1,2]. HAdV-5, as a prototype of HAdV-C, has been intensively studied and reconstructed as gene transfer vectors. Adenoviral vectors have special properties when compared with other viral vectors [3,4]. Adenoviruses have a mid-sized genome of linear double-stranded DNA which is manipulable and genetically stable. These viruses can grow efficiently in packaging cells and the virion has no envelope, which facilitates vector amplification and purification. They can infect both dividing and non-dividing cells, and many copies of the viral genome can be delivered to the nucleus of the target cell when the viruses are used at a high multiplicity of infection (MOI), resulting in high expression of the transgene. Finally, adenoviruses are non-integrating viruses and are genetically safe for the host. The main disadvantage is that adenoviruses are highly antigenic and will elicit a strong immune response against the vector in the host, which hampers re-use of the same type of adenoviral vectors in gene therapy and vaccine administration [5,6].

HAdV-5-based vectors have been widely used in basic science, gene therapy, and vaccine development. However, their application in vivo is impaired in the presence of pre-existing serum neutralizing antibodies (NAb). The seroprevalence of HAdV-5 NAb was reported to be 50–90% in healthy adults, which fluctuates slightly according to different geographic locations and ages [7,8]. Although recently HAdV-5-vectored vaccines against Ebola or SARS-CoV-2 have shown their efficacy in clinical trials, scientists are inclined to develop adenoviral vectors based on other serotypes [9,10,11,12]. Simian adenoviruses have attracted more interest because of the high biological similarity and low serum cross-reaction to their human analogs. However, efforts to develop more adenoviral vectors have been hindered by the lack of easy-to-use techniques for adenoviral genome modification.

The vector construction technique can be defined from three aspects: a technique to establish an adenovirus vector system from a wild-type virus, a technique to modify an existing vector system, or a technique used to load a transgene to an existing vector system. The principles behind these techniques can be the same, because, intrinsically, they are methods to manipulate adenovirus genomic DNA. Therefore, these techniques are not distinguished according to these applications and are non-specifically called adenovirus construction methods [13,14,15].

The homologous recombination method in bacterial cells is frequently used to construct novel adenoviral vectors. Recombinase-positive bacterial strains are an indispensable resource for this method. Such bacteria are often relatively inefficient with plasmid transformation or amplification, and some of them are commercially unavailable [16,17]. Nowadays, DNA assembly techniques are utilized to construct adenoviral vectors [11,18,19,20]. The success ratio could be low when directly assembling many long PCR products into a plasmid larger than 30 kb [20]; and it is more difficult to detect or correct point mutations introduced by PCR or DNA assembly in a plasmid of such size.

Recently, we combined the advantages of restriction enzyme digestion and Gibson assembly and simplified the procedures of introducing site-directed mutations to the adenoviral genome [21]. For convenience, we called this strategy restriction-assembly. The essence of this method is to use long restriction-digested fragments and short PCR products in a DNA assembly reaction. At the same time, the number of included DNA fragments is kept as low as possible. Restriction-assembly can be generalized as an all-purpose technique for adenovirus vector construction and modification. Here, we attempt to exemplify the versatile application of this method in the construction of E1- or E1/E3-deleted vectors based on simian adenovirus 1 (SAdV-1).

## 2. Materials and Methods

### 2.1. Cells, Virus, Plasmids and Oligonucleotides

293 (ATCC CRL-1573), HEp-2 (ATCC CCL-23), Caco-2 (ATCC no. HTB-37), VeroE6 (ATCC CRL-1586), and 293TE32 cells were cultured in Dulbecco’s Modified Eagle’s Medium (DMEM) plus 10% fetal bovine serum (FBS; HyClone, Logan, UT, USA) at 37 °C under a humidified atmosphere supplemented with 5% CO_2_, and regularly split every 3–4 days. Suspension cells of human leukemic cell lines U937 (promonocytic leukemia), K562 (chronic myelogenous leukemia), Jurkat (T-cell leukemia), and HL-60 (acute myelogenous leukemia) were cultured similarly except that RPMI-1640 medium was used instead of DMEM. 293TE32 cells were previously established by transfecting 293 cells with a eukaryotic expression plasmid carrying the HAdV-41 E1B55K gene [22].

Simian adenovirus 1 (SAdV-1; ATCC VR-195; GenBank accession No. AY771780) was amplified in 293 cells [23], and the virus genomic DNA was extracted using the modified Hirt’s method [24]. HAdV5-EG and HAdV5-GFP were E1/E3-deleted HAdV-5 vectors carrying human EF1a promoter- or CMV promoter-controlled GFP expression cassettes, respectively [25,26].

pKFAV4GFP is an adenoviral plasmid based on fowl adenovirus 4 (FAdV-4), which carries the CMV promoter-controlled GFP expression cassette [27]. pcDNA3-TF41-11p is a pcDNA3-backboned plasmid with the insertion of the HAdV-41 tripartite leader sequence (TPL) in the multiple cloning sites [22,28]. Bacteria transformation was routinely performed to screen and amplify recombinant plasmid in *Escherichia coli* TOP10 chemically competent cells with the heat shock procedure according to the manufacturer’s instructions (TIANGEN Biotech, Beijing, China).

Single-stranded DNA oligonucleotides were chemically synthesized and used as the primers for polymerase chain reaction (PCR). Information related to these oligonucleotides is summarized in Table 1.

### 2.2. Construction of E3-Deleted SAdV-1 Vector

PCR was performed to amplify the plasmid backbone containing the kanamycin resistance gene (Kan) and pBR322 origin of replication (Ori) with primer pairs of 2001KSAV1p2/p3, and the diluted PCR product was used as the template for a second round of amplification with primers of 2001KSAV1p1/p4. Such an operation helped add ITR (inverted terminal repeat) overlaps, SwaI, SalI, and AscI sites to the terminals of the PCR product. The PCR product was mixed with SAdV-1 genomic DNA and used as the substrates for the Gibson assembly reaction (NEBuilder HiFi DNA Assembly Master Mix, Cat. E2621, New England Biolabs). The product was used to transform *Escherichia coli* TOP10 competent cells. The plasmid with the Kan gene close to the E1 region was screened through restriction analysis and used for the following experiments. This plasmid (pKSAV1) contained the whole genome of SAdV-1 and was an infectious clone of the virus [29].

Overlap extension PCR was performed to amplify a 117 bp linker with primers of 2001KSAV1p5-p8. The linker was mixed with the short fragment (13,656 bp) excised from AscI-digested pKSAV1 and subjected to a DNA assembly reaction to generate an intermediate plasmid pKSAV1-AscI. PacI-E3 and E3-PmeI fragments were amplified by PCR with the template of pKSAV1-AscI; a CMV promoter-controlled GFP expression cassette of CMVp-GFP-pA (CGA) was generated by overlap-extension PCR using the pKFAV4GFP plasmid as the template (Table 1); pKSAV1-AscI was digested with PacI/PmeI and the fragment containing Kan and Ori was recovered from the agarose gel after electrophoresis; these four fragments were combined for DNA assembly to generate modified intermediate plasmid pKSAV1XE3CGA-AscI. The pKSAV1 plasmid was digested with AscI, and the larger fragment was recovered and mixed with PacI-linearized pKSAV1XE3CGA-AscI for DNA assembly to generate E3-deleted adenoviral plasmid pKSAV1XE3-CGA.

### 2.3. Construction of E1/E3-Deleted SAdV-1 Vector

Plasmid pKSAV1XE3-CGA was digested with SbfI to remove the GFP expression cassette, and the large fragment was self-ligated to generate pKSAV1XE3.

Plasmid pKSAV1XE3 was digested with SalI to recover the fragment containing the E1 region (10,528 bp), the SalI-SalI fragment inside the pTP coding sequence (CDS) was amplified by PCR, the SalI-linker was generated by self-annealing of single-stranded DNA oligonucleotides, and these three fragments were combined for DNA assembly to generate intermediate plasmid pKSAV1-SalI. Fragments of AscI-E1A and E1B-EcoRV were amplified by PCR using pKSAV1 as the template. Human EF1a promoter, GFP CDS, and BGH polyA signal (BGHpA) were generated by PCR and ligated together to generate a fragment of EF1aP-GFP-pA (EG) by overlap extension PCR. pKSAV1-SalI was digested with AscI/EcoRV to recover the fragment containing Kan-Ori, which was mixed with the other three fragments of AscI-E1A, EF1aP-GFP-pA, and E1B-EcoRV for DNA assembly to generate the modified intermediate plasmid pKSAV1XE1EG-SalI. Plasmid pKSAV1XE3 was digested with SalI, and the larger fragment was recovered and mixed with PacI-linearized pKSAV1XE1EG-SalI for DNA assembly to generate E1/E3-deleted adenoviral plasmid pKSAV1-EG0.

Extra repeats in the right ITR (RITR) and deletion in E1B19K were found by restriction analysis during the construction process and confirmed by sequencing, which originated from the genomic DNA of wild-type SAdV-1. Reverse mutation was performed to restore a normal ITR to the right end of the viral genome in pKSAV1-EG0. pKSAV1-EG0 was digested with SphI/SnaBI, diluted, and used directly as the template to amplify fragments of AsiSI-RITR and RITR-AseI (Table 1). Plasmid pKSAV1-EG0 was digested with AsiSI/AseI; the long fragment was recovered and mixed with fragments of AsiSI-RITR and RITR-AseI for DNA assembly to generate the final adenoviral plasmid pKSAV1-EG.

### 2.4. Rescue, Amplification, Purification and Titration of Recombinant Virus

The adenoviral plasmid was linearized with restriction enzyme SwaI, mixed with jetPRIME reagent, and used to transfect packaging cells according to the manufacturer’s instructions (Cat. 114-15, Polyplus-transfection, Illkirch, France). After the fluorescence foci or plaques were seen under a microscope, the cells, together with the culture medium, were subjected to three freeze–thaw rounds and spun to remove the cell debris. The supernatant was used as a seed virus for virus amplification. When the culture system scaled up to ten 15 cm dishes and after the cytopathic effect (CPE) occurred, most of the culture medium was discarded and the cells were detached with a cell lifter, collected in the remaining culture medium, and transferred to a 15 mL tube. After three freeze–thaw rounds and centrifugation, the virus in the supernatant was purified with the traditional CsCl ultracentrifugation method [30]. After dialysis in a buffer containing 10 mM Tris-Cl, 150 mM NaCl, 1 mM MgCl2, and 5% glycerol, the purified virus was aliquoted and preserved at −80 °C. Virus particle titer was determined by measuring the content of genomic DNA where 100 ng of genomic DNA is equivalent to 3.14 × 10^9^ vp (viral particles) since a 31 kb viral genome has a molecular mass of 1.92 × 10^7^. Infectious titer was determined on 293 cells with the limiting dilution assay by counting GFP+ cells 2 days post infection [31].

### 2.5. Establishment of 293 Cell Lines Stably Expressing SAdV-1 E1B55K Gene

The SAdV-1 E1B55K gene was cloned by nested PCR with primers of 2007SAV1E1Bf/r and 2007TSAV1E1Bf/r using SAdV-1 genomic DNA as the template (Table 1). The PCR product was inserted into the HindIII/XhoI site of pcDNA3-TF41-11p to generate the pcDNA3T-SAV1E1B55K plasmid by DNA assembly. pcDNA3T-SAV1E1B55K was mixed with jetPRIME reagent and used to transfect 293 cells. The cells were selected with a culture medium containing 800 μg/mL G418. G418-resistant colonies were picked out by using the cloning ring technique [32]. The cloned cells (293SE) were proliferated and preserved. Their ability in virus packaging was evaluated by observing the forming and growing of plaques after recombinant SAdV-1 infection.

The genomic DNA was extracted from 293SE cells and used as the template to amplify the SAdV-1 E1B55K expression cassette by PCR with primers of 2101SAV1E1Bs1/2 and 2101SAV1E1Bs3/4 (Table 1). The PCR products were resolved on agarose gel by electrophoresis before sequencing.

### 2.6. Transduction of Cell Lines

Adherent and suspension cell lines were infected with recombinant viruses to evaluate the gene transfer ability. For adherent cells, exponentially growing cells were seeded in 24 well plates; one day later, cells in 2 wells were detached with trypsin treatment and counted to assess the amount of cells in each well, purified viruses were diluted with culture medium and added to each well in a volume to achieve indicated MOIs, and the infection volume was adjusted to 0.25 mL/well (DMEM containing 2% FBS); the infection system was agitated once every one hour to help virus diffusion to the cell surface; after 4 h of infection, the virus-containing medium was discarded and fresh culture medium plus 2% FBS in 0.5 mL was added to each well; and 2 days post infection, the cells in each well were detached and dispersed in 0.7 mL 10 mM phosphate-buffered saline (PBS) containing 1% FBS and 1% paraformaldehyde (pH 7.4), and the expression of GFP was analyzed with flow cytometry. Suspension cells were treated similarly except that exponentially proliferating cells were counted, aliquoted, and infected directly and 0.25 mL of fresh culture medium (RPMI-1640 plus 2% FBS) was added to each well without removal of the virus after 4 h of incubation.

### 2.7. Adenovirus Neutralization Assay

Adenovirus neutralization assay was performed to determine the titer of neutralizing antibodies (NAb) in human adult sera. Serum samples were from the remnant sera of a survey program held in 2002 and were collected from 60 healthy college students aged 18–22 years in Shanxi, China. The details of this assay are described elsewhere [7]. Briefly, serum samples were 5-, 20-, 80-, and 320-fold diluted in DMEM containing 0.2% bovine serum albumin (BSA) and loaded in a 96-well plate at a volume of 50 μL per well. HAdV5-GFP or SAdV1-EG were diluted in DMEM plus 0.2% BSA to a final concentration of 4.0 × 10^7^ vp/mL or 1.6 × 10^8^ vp/mL, respectively. Diluted virus in a volume of 50 μL was mixed with the serially diluted human sera in each well. After vibration, the mixture was left to stand at room temperature for 1 h. HEp-2 cells were suspended as a single-cell solution and added to each well (2 × 10^4^ cells in 100 μL DMEM plus 4% FBS). The serum samples were finally 20-, 80-, 320-, and 1280-fold diluted because the infection volume increased from 50 μL to 200 μL after the addition of the virus and cells. The cells were cultivated for 40 h, the culture medium was removed, and the cells were fixed in 1% formaldehyde in PBS. GFP fluorescence intensity was determined with a multimode plate reader (INFINITE M1000 PRO, Tecan Austria GmbH, Grödig, Austria). The highest dilution of human serum which could inhibit more than 50% activity of the virus was defined as the NAb titer.

## 3. Results

### 3.1. Construction of E3-Deleted SAdV-1 Vector

Plasmids carrying SAdV-1 genomic DNA (pKSAV1) were constructed using a previously established method [29]. Plasmids from 12 bacterial colonies were analyzed by restriction enzyme digestion. The results showed that SAdV-1 genomic DNA was heterogenous and 4 colonies (#6, 7, 8, and 12) demonstrated the correct band pattern (Appendix A). The plasmid extracted from colony #6 was named pKSAV1 and used for the following experiments.

An intermediate plasmid (pKSAV1-AscI) was separated from pKSAV1, which carried the E3 region. The methods of PCR, restriction enzyme digestion, and Gibson assembly were used for intermediate plasmid modification and adenoviral plasmid cloning (Figure 1 and Figure 2). Recombinant virus was rescued from linearized adenoviral plasmid pKSAV1XE3-CGA-transfected 293 cells (Figure 3). The rescued SAdV1XE3-CGA had a deletion in the E3 region (XE3) and carried a CMV promoter and an SV40 polyA signal-controlled EGFP expression cassette (CMVp-GFP-pA, CGA) in the original E3 region. The viral genome was confirmed by restriction analysis (Figure 3C).

The E1, E3, E4, and ITR regions in pKSAV1 were analyzed with Sanger sequencing, and the SAdV1XE3-CGA genomic DNA was further inspected with next-generation sequencing. It was found that the cloned genome had a 660 bp deletion in the E1 region. The C-terminal of E1A and the N-terminal of E1B55K were involved, and the E1B19K coding sequence (CDS) was almost lost. The remaining E1B55K gene was in-frame with E1A CDS, which might result in the expression of an E1A-E1B55K fusion protein (Appendix A). Repeats of ITR-short (ITRs, 118 bp) and the sequence downstream of the left ITR (65 bp) were found on the right end of the cloned genome. The details are shown in Appendix A. Wild-type SAdV-1 and SAdV1XE3-CGA could grow in 293 cells, suggesting that such mutations did not inhibit or even enhanced viral replication in the HAdV-5 E1-transformed human cell line. The molecular weights in restriction analyses shown in this study were corrected according to the sequencing result.

### 3.2. Construction of E1/E3-Deleted SAdV-1 Adenoviral Plasmid

The transgene expression cassette in E3 was removed through SbfI digestion and the remaining fragment was self-ligated to generate E3-deleted plasmid pKSAV1XE3. Another intermediate plasmid (pKSAV1-SalI) was constructed for E1 modification. E1 was deleted and the human EF1a promoter and BGH polyA signal-controlled GFP expression cassette (EF1ap-GFP, EG) were inserted. Adenoviral plasmid pKSAV1-EG0 was finally generated, which was E1/E3 deleted and carried a GFP expression cassette in the original E1 region (Figure 1 and Appendix A).

SwaI-linearized pKSAV1-EG0 was used to transfect 293 or 293 TE32 cells. Small GFP foci could be found under the fluorescence microscope 7 days post transfection; they grew very slowly, and those on 293TE32 cells were relatively bigger. The rescued virus (SAdV1-EG0) was passaged once on 293TE32, and the harvested viruses were used for screening of new packaging cells (see below).

### 3.3. Establishment of SAdV-1 E1B55K-Integrated 293 Cells

293 cells constitutively express E1A and E1B55K proteins of HAdV-5. However, the function of E1B55K seems to be species specific [22,33,34], which means that E1B55K of HAdV-5 cannot complement the function of SAdV-1 E1B55K and E1-deleted SAdV-1 cannot replicate very well in 293 cells. Based on the previous experience of constructing packaging cells for HAdV-41 recombinant virus [22], we constructed SAdV-1 E1B55K eukaryotic expression plasmid pcDNA3T-SAV1E1B55K and used it to transfect 293 cells (Figure 4A). TPL in the 5′-untranslated region might enhance the expression of the target E1B55K gene and improve the yield of progeny viruses [22]. Twenty G418-resistant colonies (293SE cells) were selected, amplified, and infected with equivalent raw SAdV1-EG0 virus. Three of them (#12, 13, and 18) had superiority in virus propagation by comparing the size of the plaques and the occurrence of the cytopathic effect (CPE). Their capability in virus rescue was further evaluated by linearized pKSAV1-EG transfection (pKSAV1-EG was different from pKSAV1-EG0 in that pKSAV1-EG carried a reversely mutated ITR at the right end of the viral genome). In all three strains of cloned cells, recombinant virus SAdV1-EG could be rescued 4 days post transfection when GFP foci were seen, and rescued viruses formed plaques 6 days post transfection (Figure 4B). Comparatively, 293SE#13 grew the largest plaques. 293SE#13 (293SE13 for short) was used for the following experiments. The integration of SAdV-1 E1B55K in 293SE13 was confirmed by PCR and sequencing of the PCR product (Figure 4C), and the transcription of SAdV-1 E1B55K was detected by RT-PCR (Appendix A).

SAdV1-EG was amplified in 293SE13 cells seeded in nine 15 cm dishes and purified. Finally, a yield of 6.5 × 10^11^ vp was obtained, the physical titer was 6.4 × 10^11^ vp/mL and the biological titer was 7.0 × 10^9^ IU/mL (infectious unit per milliliter) when determined on 293 cells. The particle-to-IU ratio was close to 90. The E1/E3-deleted recombinant virus SAdV1-EG was identified by restriction analysis of the genomic DNA (Figure 4D).

### 3.4. One-Step Growth Curve of SAdV1-EG

We constructed one-step growth curves to evaluate the enhanced replication of SAdV1-EG in 293SE13 cells (Figure 5). The progeny viruses released to the culture media or associated with the cells were collected and titrated, respectively [21]. After cells were infected with a low MOI of 1 vp/cell, the progeny viruses could be detected 1-day post infection, and the virus yield increased rapidly at day 2. The virus yields experienced three waves of growth during the culture period of 10 days. It could be clearly seen that 293SE13 cells produced more progeny viruses than 293 cells at all time points. The difference in virus yields increased to more than two orders of magnitude at day 8. The interval between each growth wave was shorter in 293SE13 than in 293 cells. It was observed that most of the progeny viruses were cell associated. If the cells were infected with a high MOI of 200 vp/cell, which meant synchronized infection of more than 90% cells at the beginning, virus yield reached the peak at day 2 for both cells and decreased gradually. The virus yield in 293SE13 was about one order of magnitude higher than that in 293 cells at all indicated time points. The results illustrated that the expression of SAdV-1 E1B55K significantly improved the replication of E1/E3-deleted SAdV-1 in 293 cells.

### 3.5. Gene Transduction Efficiency of SAdV1-EG

SAdV1-EG is a novel gene transfer vector, and we evaluated its ability of gene transduction in commonly used adherent and suspension cell lines (Figure 6). For adherent cells, SAdV1-EG obtained a lower gene transduction efficiency than the control virus HAdV5-EG, which similarly carried the human EF1a promoter-controlled GFP expression cassette, or HAdV5-GFP, which used the CMV promoter. SAdV1-EG could transduce more than 90% 293 cells at an MOI of 100 vp/cell, and more than 70% of HEp-2 cells at an MOI of 500 vp/cell. Considering that HAdV-5 is well known for its high gene transduction, the capability of SAdV1-EG was acceptable. For suspension cells, SAdV1-EG was superior to HAdV-5 controls in all tested cell lines. SAdV1-EG transduced 98% of K562 cells at an MOI of 500. Interestingly, SAdV1-EG could transduce HL-60 cells efficiently. More than 70% of HL-60 cells were GFP positive when SAdV1-EG was used at an MOI of 500. As we know, HAdV-11p or HAdV-35 fiber-pseudotyped HAdV-5 could only transduce less than 10% of HL-60 cells when being used as the same MOI [26,35]. SAdV-1 contains two fiber genes in its genome and may express two types of fiber in the virion. The cellular receptors of these fibers, which to date have not been identified, are responsible for the tropism of SAdV-1 [36]. In summary, SAdV1-EG could transduce cell lines with a moderate efficiency, and it had advantages over HAdV-5 in transducing suspension cells. Notably, the transcriptional strength of CMV and EF1a promoters can be very different in hematopoietic cells [37].

### 3.6. Replication of E3-Deleted SAdV-1 in K562 Cells

Because SAdV1-EG could transduce K562 cells efficiently, it would be interesting to test if E3-deleted SAdV-1 could replicate in K562 cells. K562 cells were infected with SAdV1XE3-CGA at an MOI of 100 vp/cell for 6 h. The cells in 12-well plates were harvested every 24 h post infection, and the progeny viruses were titrated on 293 cells (because uninfected cells kept proliferating, the cells were split to avoid denutrition 3 or 6 days post infection). Transient incubation caused a portion of the cells synchronically infected, and the amount of progeny viruses reached a peak 48 h post infection. After that, the infection caused by progeny viruses was not synchronous but continuous, and the yield of progeny viruses stably increased in the following days (Figure 7A). The expression of GFP was also analyzed by flow cytometry, and it could be seen that the growth of GFP+ cells synchronized with the increase of progeny viruses (Figure 7B). These data indicated that E3-deleted SAdV-1 was replication competent, which propagated and spread in K562 cells.

### 3.7. Seroprevalence of HAdV-5 and SAdV-1 Neutralizing Antibodies in Healthy Adults

Neutralizing antibodies (NAbs) in 60 adult serum samples were titrated. NAbs against HAdV-5 were detected in 75% of these samples. In contrast, only one sample was found to possess neutralizing activity against SAdV-1, and the titer value was low (Figure 8). These results suggest that humans lack pre-existing immunity against SAdV-1.

## 4. Discussion

### 4.1. There Is a Need to Construct Novel Adenovirus Vectors

Adenoviral vectors have been extensively used in vaccine development and gene therapy. They are used for constructing human vaccines or animal vaccines to prevent the infection of many pathogens such as Ebola, influenza, SARS-CoV-2, respiratory syncytial virus (RSV), and rabies [12,13,14,38]. Oncolytic adenoviruses are armed with cell-killing or immunity-regulating genes to treat cancers [12,39]. Besides being used as vaccine vectors, helper-dependent adenoviruses are developed to treat single gene inheritance disorders [14]. The sleeping beauty transposon technique is even combined to make adenovirus an integrating vector [40,41].

However, currently available vectors cannot meet the above-mentioned needs because of the following reasons. Pre-existing immunity hampers the use or repeated use of HAdV-based vectors; adenovirus types vary in their immunogenicity ability when carrying antigen genes of pathogens, and such ability of a defined adenovirus cannot be predicted and must be evaluated through experiments [42]; adenovirus types possess different tropism features, and an adenovirus vector cannot efficiently transduce various target cells to meet the requirement of gene therapy [43]; and modifications are often necessary to load other transgenes, change tropism, or avoid pre-existing antibodies. Because of these reasons, it is urgently needed to construct adenovectors based on new serotypes or to modify existing vectors where the adenovirus vector construction techniques play their roles.

### 4.2. The Application of Restriction-Assembly in Constructing Adenovirus Vectors

Adenovirus construction techniques can be applied in three aspects, including the establishment of novel vector systems based on wild-type adenoviruses, modification of existing vector systems, and loading a transgene to form an adenovirus vector. In this study, we attempted to introduce strategies covering all these aspects (Figure 9).

First, we suggest constructing an infectious clone for the target adenovirus. An infectious clone is a handy and reliable source of the viral genome, individual viral genes, and cis-acting elements. More importantly, it can be used to evaluate if the virus rescue system works, which will provide a firm starting point for the following work. A DNA fragment of the plasmid backbone, including the replication origin and antibiotic-resistance genes, can be amplified by PCR, and this fragment carries overlaps at both ends, which originate from the 5′ terminals of the PCR primers and are homologous to the ITR ends of the target adenovirus. This PCR product is mixed with the genomic DNA of the target adenovirus, and Gibson assembly is conducted to generate a plasmid bearing the viral genome (Figure 9A). Notably, some restriction sites, which are needed for plasmid linearization or following cloning operation, can be deliberately added to the infectious clone by proper PCR primers design.

Second, an intermediate plasmid-based adenovirus vector construction procedure was proposed (Figure 9B) [21,44,45]. Mark the region to be modified in the infectious clone and carefully select unique or dual cutter restriction enzyme sites to divide the plasmid into two fragments and ensure the shorter one carries the region to be modified. If the short fragment already contains a plasmid backbone, a short linker, which carries overlaps and restriction sites for future DNA assembly or plasmid linearization, is chemically synthesized or generated by overlap-extension PCR [46]. When the short fragment does not carry a plasmid backbone, a plasmid backbone carrying overlaps and restriction sites can be amplified by PCR as described in the step of constructing the infectious clone (Figure 9A). DNA assembly can be performed to generate the intermediate plasmid. The intermediate plasmid will be much shorter than the infectious clone. More unique cutter sites can be utilized for site-directed modification by using overlap extension PCR, restriction-ligation cloning, or DNA assembly. The modified intermediate plasmid will be linearized by restriction digestion to release the modified short fragments with overlaps on both ends, which will be mixed with the large fragment from the infectious clone to generate an adenovirus vector by DNA assembly (Figure 2 and Figure 9B). Similarly, this strategy will work for modifying existing vectors if an adenovirus plasmid is used instead of an infectious clone, which was exemplified by the construction of the E1/E3-deleted adenovirus plasmid of pKSAV1-EG0 in this study (Appendix A).

Finally, the restriction-assembly can be used to load a transgene to an adenovirus vector [47]. The adenovirus plasmids constructed based on the restriction-assembly principle will contain unique or dual cutter restriction sites in the transgene cloning region. Digestion with such restriction enzymes will generate a DNA fragment with an exposed cloning site, and a transgene, which can preferably be a PCR product, will be inserted by DNA assembly (Figure 9C).

### 4.3. Advantages of Restriction-Assembly

Compared to existing methods, restriction-assembly has some advantages.

This strategy can be conveniently implemented in general molecular biology laboratories without any specialized instruments and materials. Indispensable materials include restriction enzymes, PCR reagents, Gibson assembly reagents, chemically competent *Escherichia coli* cells, bacterial culture media, and plasmid and DNA purification kits, all of which are commercially available. Experimental operation includes restriction digestion, PCR, Gibson assembly, agarose gel electrophoresis, DNA recovery, bacterial transformation, and plasmid extraction. No special professional skills are required. Especially, no recombinase-positive bacterial strain or electroporation is used in these procedures, which excludes the step of change to recombinase-negative host bacteria for efficient plasmid amplification [16,17,48]. Because of these reasons, the restriction-assembly procedure is easy to perform, cost-effective, and time saving.

Restriction-assembly is a highly effective method with a low error rate and easily verifiable outcomes. The Gibson assembly kit is designed for the ligation of 2–6 fragments [49]. In our protocol, we used these kits to ligate two fragments to generate adenoviral plasmids with a size larger than 30 kb. For the construction or modification of an intermediate plasmid, which is much shorter than adenoviral plasmid, less than four fragments are generally used. The success rate is always high because fewer fragments are included in the assembly reaction. Compared to the restriction-ligation method, restriction-assembly is always a directional cloning technique, even in cases where single-restriction digestion is employed. The robustness of this method is also manifested in that contaminated fragments, such as unseparated DNA bands from agarose gel or even digested plasmid without electrophoretic separation, can be directly used in restriction-assembly. The essence of restriction-assembly is to use long fragments from digested plasmids as much as possible. Short PCR products are included to fill gaps or to introduce site-directed mutations. As we all know, PCR can cause unpredictable mutations in the product, and unwanted point mutations can occur at the ligation sites in DNA assembly reactions. Because we always choose to use shorter PCR products and fewer fragments in DNA assembly, unwanted mutations are rarely seen. We went through the experimental notes for constructing the nine plasmids used in this study. Generally, plasmids needed to be extracted from less than 10 colonies of transformed bacteria, and the results of the restriction analysis showed that nearly all the plasmid clones were generated as pre-designed. We often selected one plasmid clone to be sequenced, and no unexpected mutation was found (Appendix A). These data indicated that restriction-assembly was highly efficient and accurate.

For the strategy we recommended, the task is broken down into steps and quality control is carried out at separated steps. Defects will be promptly detected and appropriately ameliorated. Only confirmed products are transferred to the next step, and errors will not accumulate in the final product. The construction of the SAdV-1 vector served as an example. The infectious clone was constructed at the first step, and a viable virus could be rescued from the plasmid-transfected 293 cells. That meant that the functional genome had been fixed in the plasmid, which was the prerequisite for vector construction. The intermediate plasmid was confirmed by sequencing the modified E3 region, and the E3-deleted SAdV-1 vector was verified by virus rescue. At the last step, the E3-deleted adenoviral plasmid was used to construct the E1/E3-deleted vector. In contrast, direct assembly will fuse many PCR products to generate a final adenoviral plasmid. If a recombinant virus cannot be rescued, it will be very difficult to find out where the defect comes from. The unwanted mutations can originate from PCR products, DNA assembly, or both. In this situation, you might need to sequence several clones of the whole plasmid, which is costly and time consuming. However, the problem cannot be certainly solved after sequencing because repeating experiments does not ensure an expected adenoviral plasmid without mutations in other sites. It is also possible that deletion in E1 or E3 regions leads to a changed expression of the viral genome, which might interfere with the virus replication. The researcher must determine which reason, gene mutations or improper experimental design, results in the failure of rescuing recombinant virus.

Vector systems established by restriction-assembly meet the requirements of sustainable development. First, intermediate plasmids are useful components of the system. For example, in this study, the intermediate plasmid pKSAV1-AscI was generated to construct pKSAV1XE3CGA-AscI for modification of the SAdV-1 E3 region (Figure 2). After pKSAV1XE3-CGA was successfully constructed, did pKSAV1-AscI or pKSAV1XE3CGA-AscI become useless? Far from that, they could still be used for the modification of any gene or cis-acting elements in them, such as fiber or the genes in the E4 region. The modification could be conveniently transferred into the SAdV-1 genome by restriction-assembly, just as demonstrated in E3 modification. In contrast, a shuttle plasmid in other adenovirus vector systems is constructed for single gene transfer and is often used just once. Second, the adenoviral plasmids are prepared for transgene replacement or further modification. For example, a transgene or transgene expression cassette can be directly inserted to the SpeI or FseI sites in pKSAV1-EG to generate new E1/E3-deleted vectors by restriction-assembly. Similarly, gene elements can also be transferred into the E3 region by using the SbfI site. Interestingly, we can envision an oncolytic SAdV-1 that is generated by restoring tumor-specific promoter-controlled E1 genes to the FseI site in pKSAV1-EG. As mentioned earlier, the E1/E3/E4 genes can be easily modified by using the intermediate plasmids. Obviously, modification of other parts in the genome, such as penton and hexon, can also be realized through constructing other intermediate plasmids according to the principle of restriction-assembly. On the contrary, the adenovirus plasmids constructed by other methods are often dead ends and can only be used to rescue recombinant viruses.

Based on the above analysis, it can be seen that restriction-assembly is an ideal strategy for the synthetic biology of adenoviruses [50]. Unique or dual restriction enzyme cut sites are selected from or artificially placed on an adenoviral plasmid. They provide entrances or access points in the viral genome for replacement or insertion of DNA fragments by restriction-assembly. The DNA fragments can be PCR products or from digested intermediate plasmids. Modularity is the most important feature of synthetic biology. The modularity of the adenovirus vector can be realized if the principle of restriction-assembly is applied and adhered to. Taking the SAdV-1 vector system as an example, the transgene module can be assembled to the E1 region (SpeI or FseI sites) or E3 region (SbfI site), while the modification of E3/E4 regions can be implemented with the help of intermediate plasmid pKSAV1-AscI at the AscI-AscI module (the short fragment between dual AscI sites in the adenoviral plasmid). It can be expected that the functions of the system will be enhanced, specialized, or expanded by continuous improvement.

Conclusively, based on others’ work and our experience [11,21,27,44,47], we present restriction-assembly as an all-purpose method for the construction of novel adenovirus vectors or modification of existing adenovector systems. This strategy possesses many advantages and will benefit the synthetic biology of adenoviruses.

## 5. Patents

A patent application has been filed on this work by the National Institute for Viral Disease Control and Prevention, Chinese Center for Disease Control and Prevention.

## Figures and Tables

**Figure 1 viruses-14-00546-f001:**
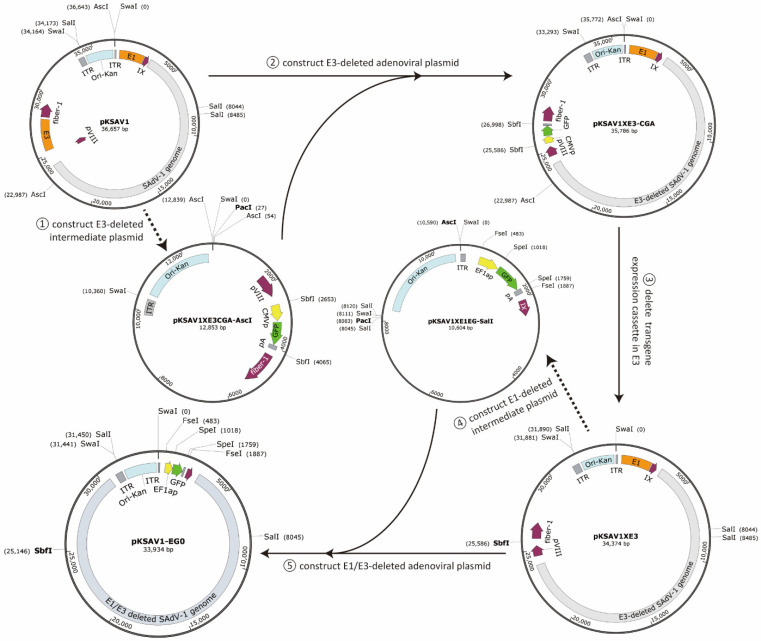
Schematic diagram for constructing E3-deleted and E1/E3-deleted SAdV-1 adenoviral vectors. The construction procedure was divided into 3 steps: separate a small intermediate plasmid from the large adenoviral plasmid; modify the intermediate plasmid by deleting some viral genes and inserting transgenes; and restore the modified intermediate plasmid to the original plasmid to generate final adenoviral plasmids. Because the DNA fragments were mainly generated by restriction enzyme digestion and Gibson assembly reagent instead of DNA ligase was routinely used in our procedure, we called this method restriction-assembly for convenience. The adenoviral plasmids, such as pKSAV1XE3-CGA and pKSAV1-CG0, can be linearized with SwaI digestion and used to transfect packaging cells for recombinant virus rescue. On the other hand, the adenoviral plasmid is the starting material for constructing recombinant vectors carrying other transgenes: the existing transgene can be removed by restriction digestion (SbfI, SpeI, or FseI in this case), and the new transgene can be inserted by restriction-ligation cloning or restriction-assembly. Generally, a shuttle plasmid is dispensable, a transgene fragment can be amplified by PCR, and only one adenoviral plasmid is needed for vector construction, so we called it single plasmid-based adenoviral vector system.

**Figure 2 viruses-14-00546-f002:**
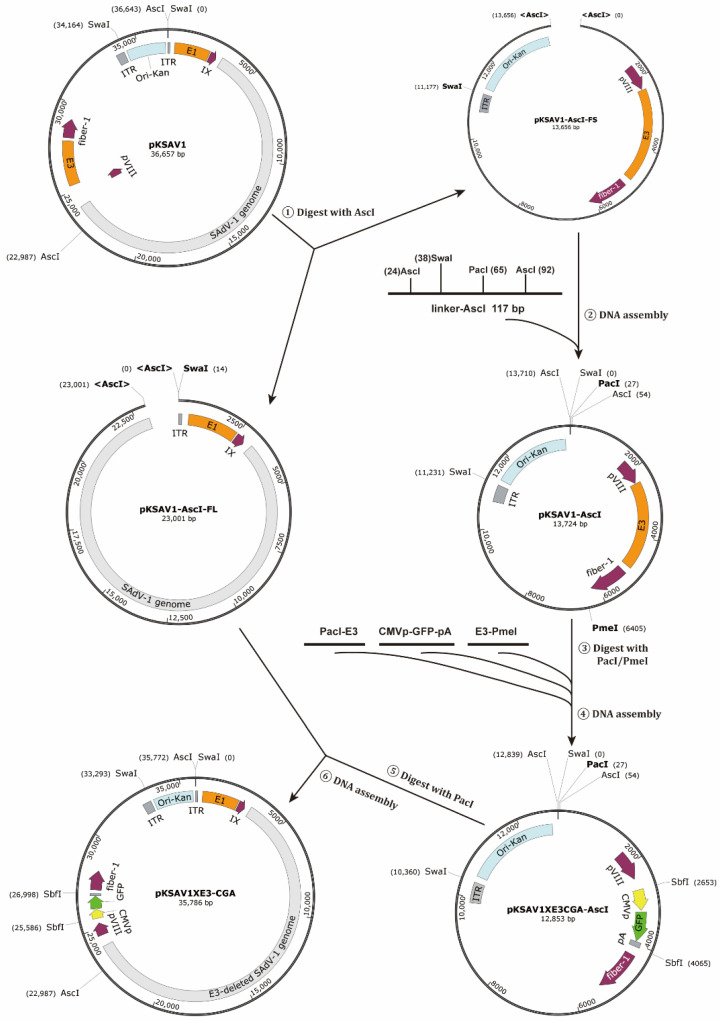
Schematic diagram of constructing adenoviral plasmid pKSAV1XE3-CGA.

**Figure 3 viruses-14-00546-f003:**
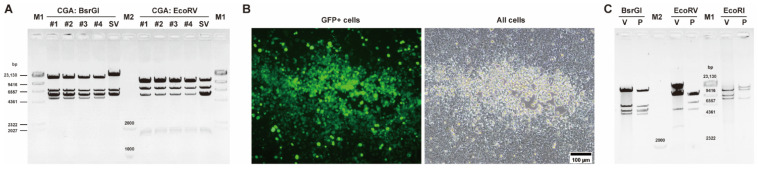
Rescue and identification of E3-deleted SAdV-1 vector. (**A**) Restriction analysis of E3-deleted adenoviral plasmid pKSAV1XE3-CGA (CGA). Plasmids were extracted from 4 bacterial colonies (#1–#4). Plasmid pKSAV1 (SV) was used as a control. The predicted molecular weights (bp) of digested fragments of CGA were 4797, 5313, 5541, 6428, and 13,707 for BsrGI; and 564, 1639, 1732, 5245, 6868, 9542 and 10,196 for EcoRV. The predicted molecular weights (bp) of digested fragments of SV were 5313, 5541, 6428, and 19,375 for BsrGI; and 564, 1639, 1732, 5178, 5235, 5245, 6868, and 10,196 for EcoRV. (**B**) Rescue of recombinant simian adenovirus SAdV1XE3-CGA in 293 cells. SwaI-linearized pKSAV1XE3-CGA was used to transfect 293 cells, and foci formed by GFP-positive cells could be observed under a fluorescence microscope 7 days post transfection. (**C**) Identification of SAdV1XE3-CGA by restriction analysis of the viral genomic DNA. The predicted molecular weights (bp) of digested fragments of virus genomic DNA (V) were 1392, 1648, 4797, 5313, 6428, and 13,707 for BsrGI; 564, 1504, 1639, 1732, 2863, 5245, 9542, and 10,196 for EcoRV; and 3613, 8038, 9222 and 12,412 for EcoRI. The predicted molecular weights (bp) of digested fragments of pKSAV1XE3-CGA plasmid (P) were 8038, 12,412, and 15,336 for EcoRI.

**Figure 4 viruses-14-00546-f004:**
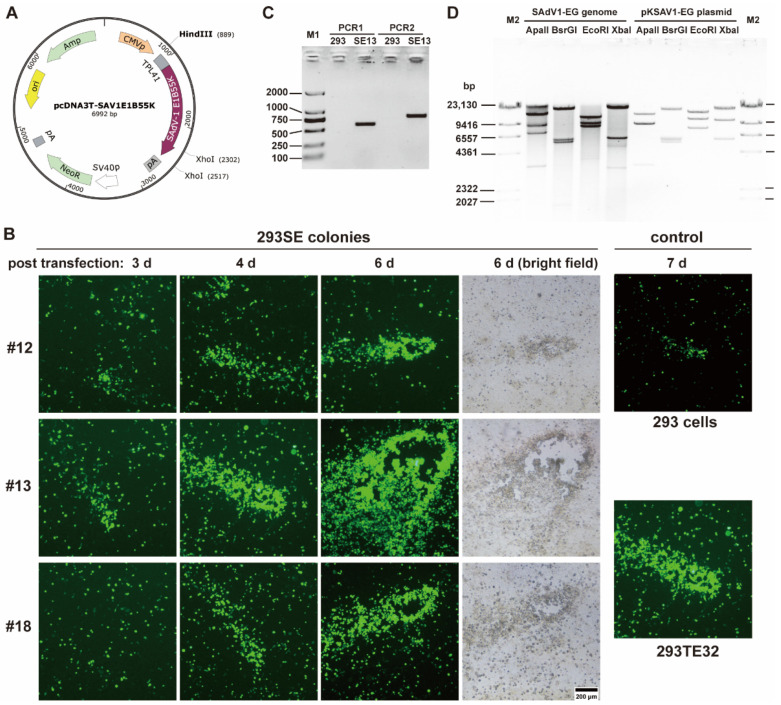
Establishment of SAdV-1 E1B55K-integrated cell lines for recombinant virus packaging. (**A**) Plasmid map of pcDNA3T-SAV1E1B55K. HAdV-41 tripartite leader sequence (TPL41) and SAdV-1 E1B55K coding sequence (CDS) were inserted into the multiple cloning site of pcDNA3 eukaryotic expression plasmid (between HindIII/XhoI sites). CMVp: CMV promoter; pA: polyA signal. (**B**) Evaluation of the ability of selected cell colonies for virus packaging. Twenty G418-resistant pcDNA3T-SAV1E1B55K-transfected 293 cell colonies were firstly screened by observing the formation of plaques and cytopathic effect (CPE) after being infected with SAdV1-EG0 virus. Three colonies (293SE#12, 13, and 18) could grow more and bigger plaques and demonstrated CPE more rapidly. The virus rescue and propagation ability were further assessed by transfecting linearized pKSAV1-EG into these 3 strains of cloned cells. 293 and 293TE32 cells were used as controls. The results of the transfection experiments are shown. Small GFP+ foci could be seen on 293SE#12 and #13 cells 3 days post transfection. Plaques were found on all colonies while relatively bigger plaques were formed on 293SE#13. Small GFP foci but no plaques were observed on control cells. 293SE#13 (293SE13 for short) was used for the following experiments. (**C**) Identification of the integration of SAdV-1 E1B55K in 293SE13 cells. After 293SE13 cells were passaged 16 times, genomic DNA was extracted and used as the template for amplifying SAdV-1 E1B55K expression cassette with primers located inside the CMV promoter and CDS region. PCR products with the expected molecular weight were recovered from agarose gel after electrophoresis and further confirmed by sequencing. No corresponding bands could be amplified when using genomic DNA of 293 cells as the template. (**D**) Identification of SAdV1-EG virus. Virus genomic DNA was extracted from SAdV1-EG-infected 293SE13 cells and digested with restriction enzymes. The products were resolved on agarose gel. Plasmid DNA of pKSAV1-EG was treated similarly and served as a control.

**Figure 5 viruses-14-00546-f005:**
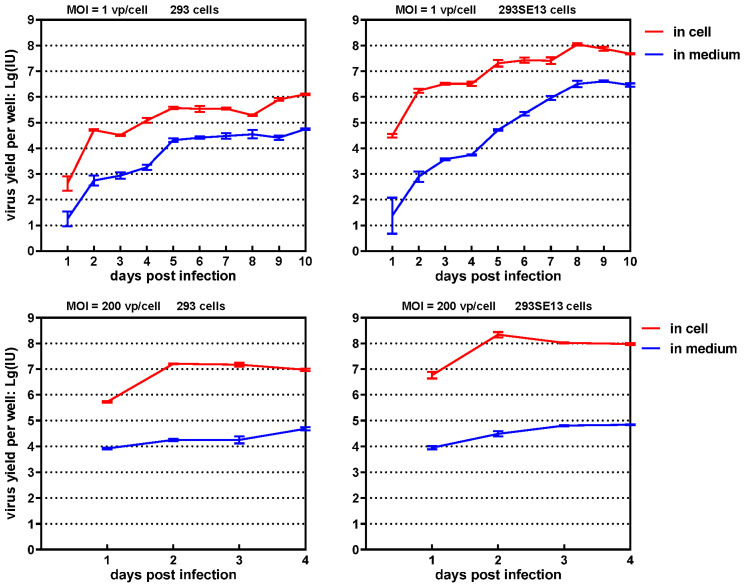
One-step growth curves of SAdV1-EG in 293 or 293SE13 cells. Cells were seeded in 12-well plates and infected with purified SAdV1-EG at an MOI of 1 or 200 vp/cell for 2 h. Cells and the culture media were collected separately at indicated time points post infection. After titration, the yields in each well were calculated and used to construct the growth curves.

**Figure 6 viruses-14-00546-f006:**
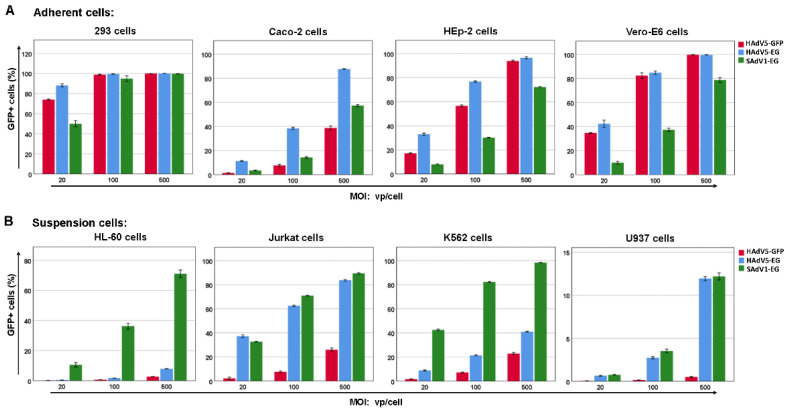
Gene transduction efficiency of SAdV1-EG on adherent (**A**) or suspension cell lines (**B**). Cells were infected with SAdV1-EG at an MOI of 20, 100, or 500 vp/cell. The percentage of GFP+ cells was determined by flow cytometry 2 days post infection. HAdV5-GFP and HAdV5-EG viruses served as controls, which were E1/E3-deleted HAdV-5 vectors and carried CMV promoter (HAdV5-GFP) or human EF1a promoter (HAdV5-EG) controlled GFP cassette, respectively.

**Figure 7 viruses-14-00546-f007:**
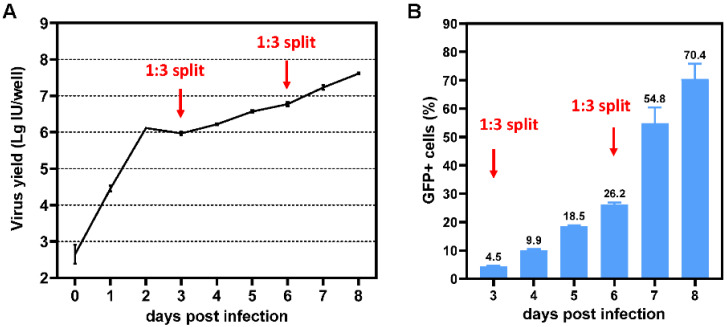
Infection of K562 cells by E3-deleted SAdV-1 vector. K562 cells were infected by SAdV1XE3-CGA at an MOI of 100 vp/cell for 6 h. The unadsorbed viruses in the media were removed by centrifugation, and the cells were washed once, suspended in RPMI-1640 media plus 2% FBS, and aliquoted into each well of 12-well plates. Three or 6 days post infection, 2 mL fresh culture media was added to each well to reach a final volume of 3 mL, and the cells were suspended by aspiration and aliquoted into 3 wells of new 12-well plates to avoid denutrition. At each time point post infection, cells together with the culture media were harvested, subjected to 3 freeze–thaw rounds, and centrifuged to remove cellular debris, and the viruses in the supernatant were titrated on 293 cells (**A**). Cells were also collected to analyze GFP expression by flow cytometry assay (**B**).

**Figure 8 viruses-14-00546-f008:**
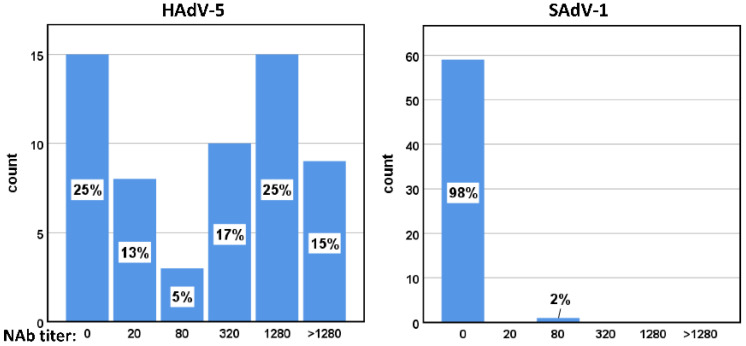
Overall distributions of adult serum samples according to neutralizing antibody (NAb) titers (*n* = 60). Human sera were serially diluted, and the highest dilution of serum which could inhibit more than 50% activity of the virus was defined as the NAb titer. A titer of <20 was defined as negative. The y-axis represented the number of samples, while the proportions of samples with indicated NAb titer were shown inside each bar.

**Figure 9 viruses-14-00546-f009:**
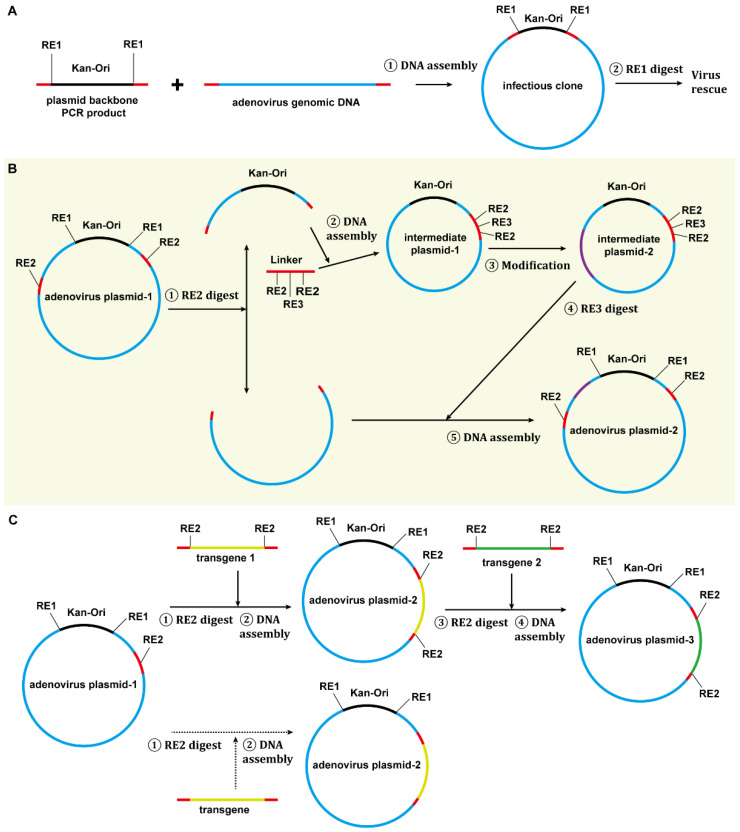
Strategies for adenovirus vector construction and modification. (**A**) Construction of an adenovirus infectious clone. (**B**) Construction or modification of a novel adenovirus vector system by the method of restriction-assembly. Intrinsically, it is an intermediate plasmid-based strategy to modify adenoviral genome where restriction-assembly simplifies the construction procedure. When adenovirus plasmid-1 is the infectious clone of a new serotype adenovirus, a novel vector system can be established by this strategy. If adenovirus plasmid-1 belongs to an existing vector system, it will generate a modified one. (**C**) Load a transgene to an adenovirus vector using the restriction-assembly method. The prerequisite of this procedure is that the start adenovirus plasmid (adenovirus plasmid-1) is constructed in advance to satisfy the requirement of restriction-assembly: a unique cutter site (RE2) is placed as the entrance for insertion of exogenous DNA. If RE2 sites are saved to flank the transgene-1, the product plasmid (adenovirus plasmid-2) can be re-used for loading other transgenes just as adenovirus plasmid-1 does. The RE2 site can be removed, if necessary, as shown by the dotted branch. Overlaps for Gibson assembly are shown in red. Kan-Ori: kanamycin-resistance gene and pBR322 origin of replication (plasmid backbone); RE: restriction enzyme.

**Table 1 viruses-14-00546-t001:** Summary information of oligonucleotides.

Fragment	Oligo Name	Sequence	Template	Product (bp)	Restriction Enzyme
Kan-Ori	2001KSAV1p1	gtttccagaa taaggtatat tattgatgat gatttaaatc caagtcgac	pKFAV4GFP	2514	SwaI, SalI
	2001KSAV1p2	tgatgatgat ttaaatccaa gtcgacgatc ccgagcggta tcagctc			
	2001KSAV1p3	tgatgattta aatggttggc gcgcctggaa caacactcaa ccctatcg		2563	
	2001KSAV1p4	gtttccagaa taaggtatat tattgatgat gatttaaatg gttggcgcgc c			SwaI, AscI
AscI-linker	2001KSAV1p5	gatagggttg agtgttgttc caggcgcgcc aaccatttaa atcatcat		117	AscI, SwaI
	2001KSAV1p6	gcgcgccaac catttaaatc atcatcaata atatacctta attaagac	self-anneal		SwaI, PacI
	2001KSAV1p7	cgccgctggc ggcagaggag tttgtcttaa ttaaggtata ttattgat			PacI
	2001KSAV1p8	cgctgaaacc ggaccacagg gcgcgccgct ggcggcagag gagtttgt			AscI
PacI-E3	2001KSAV1p9	cgcgccaacc atttaaatca tcatcaataa tatacctt	pKSAV1-ASCI	2680	
	2001KSAV1p10	gttatgtaac gcctgcagga tgtaatccgg gcgtggggca g			SbfI
CMVp-GFP-pA	2001KSAV1p11	ggattacatc ctgcaggcgt tacataactt acggtaaatg	pKFAV4GFP	568	SbfI
	2001KSAV1p12	accatggtgg ctagctctag cggatctgac ggttcactaa ac			
	2001KSAV1p13	gatccgctag agctagccac catggtgagc aagggcg		770	
	2001KSAV1p14	caataaacaa gttagctagc ttagagtccg gacttgtaca gctcgtcc			
	2001KSAV1p15	gtccggactc taagctagct aacttgttta ttgcagctta taatggttac		163	
	2001KSAV1p16	ccttaaaaat atccctgcag gtaagataca ttgatgagtt tggacaaacc ac		1442	SbfI
E3-PmeI	2001KSAV1p17	catcaatgta tcttacctgc agggatattt ttaaggtgta aatcaataat aaacttacc	pKSAV1-ASCI	1522	HindIII
	2001KSAV1p18	cattttgcgt agtaatggga tctctgtagt ttaagcttaa cactccaagt gg			
SalI-SalI	2001KSAV1p19	gacgctccat ggcctcgtag aagtccacgg cgaagttgaa aaattg	pKSAV1	529	PacI
	2001KSAV1p20	aatcatcatc aataatatac cttattaatt aacgctttcc tagagaagtt ctcggatc			
SalI-linker	2001KSAV1p21	gcgttaatta ataaggtata ttattgatga tgatttaaat ccaagtcgac	self-anneal	73	
	2001KSAV1p22	gtgagctgat accgctcggg atcgtcgact tggatttaaa tcatcatcaa			
AscI-E1A	2001KSAV1p23	cgcgataggg ttgagtgttg ttccagg	pKSAV1	530	
	2001KSAV1p24	gagcggccgg cccgcggcag cgcggaggag aaaac			FseI
EF1ap-GFP-pA	2001KSAV1p25	cgctgccgcg ggccggccgc tccggtgccc gtcagtggg	pCDH-CMV-MCS-EF1-copGFP	566	FseI
	2001KSAV1p26	catggtggc actagtgtag gcgccggtca c			SpeI
	2003SAV1EGFPf	gtgaccggcg cctacactag tgccaccatg gtgagcaagg g	pKSAV1E3CGA	794	SpeI
	2003SAV2EGFPr	ggtcaaggaa ggcacggggg agactagttt agagtccgga cttgtacagc tc			SpeI
	2001KSAV1p29	gactctaaac tagtctcccc cgtgccttcc ttgacc	pcDNA3	151	SpeI
	2001KSAV1p30	ggatacaacc tcggccggcc accccacccc ccagaataga atg			FseI
E1B-EcoRV	2001KSAV1p31	ggtggccggc cgaggttgta tcctgtaacc ctgaacgt	pKSAV1	579	FseI
	2001KSAV1p32	tctgaagcgg tatcggggtt agcttgggat			
AsiSI-RITR	2010SAdV1RITR1	taacagaccc aggtcaggtt gctctc	pKSAV1-EG0/SphI-SnaBI	997	
	2010SAdV1RITR2	cgagggcggg cgggcgaagg gcgtgtgtgg gaaag			
RITR-AseI	2010SAdV1RITR3	ccttcgcccg cccgccctcg cgccaccccg cgtca	pKSAV1-EG0/SphI-SnaBI	1324	
	2010SAdV1RITR4	cgaactactt actctagctt cccggcaaca			
SAdV1-E1B55K	2007SAV1E1Bf	gctgccttta ttacctatat tttgg	SAdV-1 genome	1489	
	2007SAV1E1Br	cctcatgccc ctttataccc tt			
	2007TSAV1E1Bf	tcgagccaat cacagtcgca agatggagca acagcgacag cc		1466	
	2007TSAV1E1Br	agggccctct agatgcatgc tcgagtcact cctcatcgct ggattcat			
CMVp-TPL	2101SAV1E1Bs1	gatagcggtt tgactcacgg	293SE13 genome	594	
	2101SAV1E1Bs2	gttctcctcc accactcggt			
SAV1E1B55K	2101SAV1E1Bs3	tggagcaaca gcgacagcc	293SE13 genome	811	
	2101SAV1E1Bs4	accgccttcc agcaaccat			

## Data Availability

Not applicable.

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
