# Peer review of "Restriction-Assembly: A Solution to Construct Novel Adenovirus Vector"

_viruses, 2022, doi:10.3390/v14030546_

Round 1
Reviewer 1 Report
This version is improved and could be published after minor adjustments
Author Response
Response to the first reviewer expert:
The professor had no negative comments. We modified the manuscript according to the suggestions from the second reviewer.
Reviewer 2 Report
Manuscript by Gou et al deals with novel cloning approach of adenovirus vectors. Their developed method combines both classical restriction enzyme-based assembly with more recent Gibson assembly. The method is extremely well described and results are nicely documented. The cherry on the cake is elegant figure 9, which takes the whole strategy together. Also one has to give a good credit for the discussion part where the authors uplift the advantages of their developed method.
Although the manuscript has a good quality and interesting technical approach, I have few suggestions how improve the quality of the manuscript. Hence, I hope that the authors will really consider my suggestions and remarks.
Abstract (L23-27): I strongly suggest to shortern the abstract and eliminate too detailed descriptions of the cloning strategies. This part would rather belong to the M&M section and as it is now it repels the reader like me as it is to technical abstract.
Intro:
L43: replace genotypes with types as that reflects better to the previous and current classifications.
L47-56: This has to be the worlds longest sentence! To make it readable, make at least 3-4 short sentences of it and make them in a way that they make logical sense.
L76: unfriendly is not proper word here, replace it
Section 3.3: although the reason why 293SE cell line was made is provided, it would be more informative to explain why tripartite leader sequence was used in the E1B55K plasmid. Do the authors have experiences that E1b55K expression is lower w/o Trip sequence? Also in Figure 4, it would be elegant to show western blot with anti-E1B55K antibody. Unless the SAdV-1 E1B55K is not recognized with classical 2A6 antibody. I am also aware that 2A6 will recognize the HAdV-5 E1B55K, but still the Simian version should be detected in the stable cell line 293SE.
Author Response
Response to the second reviewer expert:
- Abstract (L23-27): I strongly suggest to shortern the abstract and eliminate too detailed descriptions of the cloning strategies. This part would rather belong to the M&M section and as it is now it repels the reader like me as it is to technical abstract.
We really wish that the professor will allow us to keep these 2 sentences in the abstract. The topic of our manuscript is to introduce a strategy for adenovirus vector construction. We think that some readers might welcome a concise description of the construction procedure in the abstract.
- L43: replace genotypes with types as that reflects better to the previous and current classifications.
We changed “genotypes” to “types” as the professor suggested. Thanks.
- L47-56: This has to be the worlds longest sentence! To make it readable, make at least 3-4 short sentences of it and make them in a way that they make logical sense.
We split the long sentence into 5 short ones as the professor suggested. Thanks.
- L76: unfriendly is not proper word here, replace it.
We changed “unfriendly” to “relatively inefficient”.
- Section 3.3: although the reason why 293SE cell line was made is provided, it would be more informative to explain why tripartite leader sequence was used in the E1B55K plasmid. Do the authors have experiences that E1b55K expression is lower w/o Trip sequence?
Yes, we do have such experiences. We used to package HAdV-41 vector in HAdV-41 E1B55K-expressing 293 cells. If TPL was included in the eukaryotic expression plasmid, the expression of HAdV-41 E1B55K would be improved and the yield of progeny viruses would go up. To follow the professor’s suggestion, we added one more sentence to explain in “3.3. Establishment of SAdV-1 E1B55K-integrated 293 cells” in the Results section.
line 311-312:
TPL in the 5'-untranslated region might enhance the expression of target E1B55K gene and improve the yield of progeny viruses [22].
- Also in Figure 4, it would be elegant to show western blot with anti-E1B55K antibody. Unless the SAdV-1 E1B55K is not recognized with classical 2A6 antibody. I am also aware that 2A6 will recognize the HAdV-5 E1B55K, but still the Simian version should be detected in the stable cell line 293SE.
The similarities between SAdV-1 E1B55K and HAdV-5 E1B55K were attached below. 293 cells constitutively express HAdV-5 E1B55K, while 293SE13 cells simultaneously express HAdV-5 E1B55K and SAdV-1 E1B55K. We don’t know if commercial monoclonal antibody against HAdV-5 E1B55K could recognize SAdV-1 E1B55K, and we don’t have these antibodies at hand. After reading the professor’s comments, we performed the Western blot by using the mouse antiserum against truncated HAdV-41 E1B55K, which were previously prepared in the laboratory (J Virol Methods. 2011; 175:188-96. PMID: 21601594). However, the results were unexpected: specific band with a molecular weight about 40 kd were detected on both 293SE13 and 293 protein loading lanes. It was possible that this antiserum could not efficiently bind HAdV-5 and SAdV-1 E1B55K, and instead, it recognized an unrelated protein or digested E1B55K (data not shown).
Considering there was no appropriate antibody to specifically detect SAdV-1 E1B55K, we performed RT-PCR to determine the transcription of SAdV-1 E1B55K in 293SE13 cells. The results were summarized in Figure S4, and one sentence was added to the end of 3.3. Establishment of SAdV-1 E1B55K-integrated 293 cells (line 323-324).
, and the transcription of SAdV-1 E1B55K was detected by RT-PCR (Figure S4).
Figure S4. Detect the transcription of SAdV-1 E1B55K in 293SE13 cells by semi-quantity PCR.
293 or 293SE13 (SE13) cells were lysed in TRIzol reagent (Thermo Fisher, Cat. 15596026). The RNA was extracted, treated with DNaseI (Thermo Fisher, Cat. 18068-015), reversed transcribed to cDNA with oligo(dT)18 as the primer (PrimeScript II 1st Strand cDNA Synthesis Kit, Cat. 6210A; Takara), 1:40 diluted in water and used as the PCR templates. Semi-quantitative PCR was employed to detect the transcription of SAdV-1 E1B55K with primers of 2101SAV1E1Bs3 and 2101SAV1E1Bs4 (Table 1), and the transcription of human b-actin gene (GenBank NM_001101) was also determined and served as a template loading control [1]. The primers for human b-actin were 1304actin-F (tggcacccagcacaatgaa) and 1304actin-R (ctaagtcatagtccgcctagaagca). M: DL2000 DNA marker (Takara).
Reference:
[1] Kinoshita, T., Imamura, J., Nagai, H., Shimotohno, K., 1992. Quantification of gene expression over a wide range by the polymerase chain reaction. Anal. Biochem. 206 (2), 231–235.

This manuscript is a resubmission of an earlier submission. The following is a list of the peer review reports and author responses from that submission.
Round 1
Reviewer 1 Report
Corresponding author: Zhuozhuang Lu
Title: Restriction-Assembly: A solution to construct Novel Adenovirus Vector.
General comments:
In this manuscript, Lu et al., generated E1- or E1/E3-deleted vectors based on simian adenovirus 1 (SAdV-1) using restriction-assembly strategy. Their findings suggest that restriction-assembly can simplify the procedures of introducing site-directed mutations to adenoviral genome through the combination restriction enzyme digestion and Gibson assembly. Although some of their findings are interesting, authors make premature conclusions without corresponding in-depth mechanism analysis being presented in their manuscript. Due to these shortcomings, the manuscript is not suitable for publication in current state.
Specific comments:
- In general, authors stated that the restriction-assembly holds many advantages compared to conventional vector construction technique. However, authors have not shown any data proving this point such as comparing the success rate of constructing adenoviral vectors and the transfection efficiency of the vectors to produce virus. Moreover, the restriction-assembly technique shows similar features as the infusion cloning technique that is being used commercially. Therefore, authors should provide more explanation and data to prove the superiority of restriction-assembly.
- Throughout the manuscript, authors wrote “SAdV1-EG0” and “SAdV1-EG” to represent the E1/E3-deleted SAdV-1 virus. Since both designation means the same thing, authors should use only one name for the virus.
- In Figure 4, authors used 293TE32 as a control to show that the construction of a new packaging cell expressing SAdV-1 E1B55K could increase transfection efficiency. However, there are no explanation on what the 293TE32 cell is. Therefore, authors should provide details of 293TE32 cell.
- Regarding Figure 5, authors described that when cells were infected with low MOI of 1 vp/cell, the virus yield was steady in 293 cells starting from day 2 while 293SE13 cells showed increase in virus yield until day 8. However, with such low MOI, adenovirus should show increase in virus yield even after day 2. Therefore, authors should provide more detailed explanation of this phenomenon.
- In Figure 6, authors evaluated SAdV1-EG’s gene transduction efficiency in adherent cells and suspension cells. All suspension cells are human leukemic cells, which are mostly known for low coxsackie adenovirus receptor (CAR) expression. However, based on the result, HAdV5-EG and SAdV1-EG has high transduction efficiency even though there has been no modification in the fiber. Moreover, concerning HAdV5-GFP and HAdV5-EG, which have the same fiber, there is a big difference in transduction efficiency due to different promoters. Authors must provide in depth analysis for the reason of high transduction efficiency in low CAR cells and why having different promoters cause such big difference in transduction efficiency.
- According to Figure 7, authors provide data concerning the replication of E3-deleted SAdV-1 in K562 cells. In this experiment, authors stated that 1:3 split was made on day 3 and 6; however, they failed to mention the reason for the 1:3 split. Moreover, if the virus infected cells were split into 3 wells, how could the virus yield or the GFP expression be higher on day 4 and day 7. Authors should provide detail on the purpose of the 1:3 split of the virus infected cells and why it was able to have higher virus yield after the 1:3 split.
Reviewer 2 Report
In this study, the authors outline a cloning strategy using restriction enzyme digestion on overlapping segments combined with gibson assembly for the production of novel adenoviral genomes. They use this strategy to build a simian adenovirus genome. They show infectivity of this virus as compared to HAdV5. These simian adenovirus genomes do not have the already present antibody response that exists for human adenovirus and so they may be better suited for gene therapy. Current strategies for making novel adenovirus are cumbersome and this paper outlines a more streamlined approach.
Are there more than 100 Human adenoviruses? Please reference this.
What are the limits to capacity for simian adenovirus?
HAdV5 is often used at an MOI of 5 or 10 with high infectivity. Please explain the discrepancy in infectivity in figure 6.
This strategy is good for making novel adenovirus. Once a clone is established, how would this strategy compare with CRE adenovirus? Would these strategies be complementary (i.e. would you build CRE into this strategy)?
Reviewer 3 Report
I read your manuscript with great interest, an additional complementary method of generating viral mutants. The manuscript describes all methods reasonably well but could have been more detailed to encourage other scientist to use the method.
The method the authors describe is basically a combination of previous restriction enzyme cloning, and PCR fragment amplification with Gibson assembly. This method although on a smaller scale is already used by other researchers and is not that novel. The novelty of this manuscript is the extensive generation of plasmids that can be used as tools in generation of several mutants.
I find the manuscript difficult to read and understand, mostly because of the language that need to be improved; grammar and choice of words makes some sections impossible to understand. More specific and detailed descriptions would also be required.
Some issues that need to be addressed:
Why does figure 6 not include the 293SE13 cells? Would be very informative to compare. Do these cells also support Ad5 to same degree?
Figure 7 need further explanation as to split and relevance of this.
Figure 8 needs explanations as to what is measured on the y-axes and what the % of activity refer to. Figure may not be relevant to the main message of the study?
The discussion is very long and is trying to explain why the authors' method is better than other viral engineering methods. However, for this reviewer I find the points being slightly lost in the rather superficial explanations. Would be good to have more side by side scientitic relevant comparisons. For example, what are success rates and how long will each procedure typically take?
Round 2
Reviewer 1 Report
Corresponding author: Zhuozhuang Lu
Title: Restriction-Assembly: A solution to construct Novel Adenovirus Vector.
Specific comments:
- In the first round of revision, we requested authors to provide data to prove the superiority of restriction-assembly. Authors provided explanation stating that “the occurrence of unwanted mutations is at a lower level when compared with direct assembly of many PCR products”. However, authors have not presented any new data concerning the comparison of infusion cloning, homologous recombination, and restriction assembly. Authors should at least compare the cloning accuracy and number of negative control colonies using each cloning method to clearly show that restriction-assembly is superior in certain aspects.
- In Figure 5, authors stated that when cells were infected with low MOI of 1 vp/cell, virus replication is not strong enough to lyse the host cell and release the progeny viruses. Therefore, the culture preserves a constant amount of progeny virus in the following days. However, this is only a theory, and no scientific data has been presented to support their point.
- In Figure 6, we requested in depth analysis for the reason of high transduction efficiency in low CAR expressing cells. However, authors provided no explanation concerning this point.
